# Coordination and Cognition in Pure Nutritional Wernicke’s Encephalopathy with Cerebellar Degeneration after COVID-19 Infection: A Unique Case Report

**DOI:** 10.3390/jcm12072511

**Published:** 2023-03-27

**Authors:** Nicolaas J. M. Arts, Maud E. G. van Dorst, Sandra H. Vos, Roy P. C. Kessels

**Affiliations:** 1Centre of Excellence for Korsakoff and Alcohol-Related Cognitive Disorders, Vincent van Gogh Institute for Psychiatry, 5803 DN Venray, The Netherlands; 2Winkler Neuropsychiatry Clinic and Korsakoff Centre, Pro Persona Institute for Psychiatry, 6874 BE Wolfheze, The Netherlands; 3Donders Institute for Brain, Cognition and Behaviour, Radboud University, 6525 GD Nijmegen, The Netherlands; 4Tactus Addiction Care, 7400 AD Deventer, The Netherlands

**Keywords:** alcoholic cerebellar degeneration, nutritional cerebellar degeneration, Wernicke’s encephalopathy, cerebellar atrophy, cerebellar disfunction, neuropsychology, Korsakoff’s syndrome, COVID-19, thiamine deficiency, cerebellar cognitive affective syndrome

## Abstract

Background: Alcoholic cerebellar degeneration is a restricted form of cerebellar degeneration, clinically leading to an ataxia of stance and gait and occurring in the context of alcohol misuse in combination with malnutrition and thiamine depletion. However, a similar degeneration may also develop after non-alcoholic malnutrition, but evidence for a lasting ataxia of stance and gait and lasting abnormalities in the cerebellum is lacking in the few patients described with purely nutritional cerebellar degeneration (NCD). Methods: We present a case of a 46-year-old woman who developed NCD and Wernicke’s encephalopathy (WE) due to COVID-19 and protracted vomiting, resulting in thiamine depletion. We present her clinical course over the first 6 months after the diagnosis of NCD and WE, with thorough neuropsychological and neurological examinations, standardized clinical observations, laboratory investigations, and repeated MRIs. Results: We found a persistent ataxia of stance and gait and evidence for an irreversible restricted cerebellar degeneration. However, the initial cognitive impairments resolved. Conclusions: Our study shows that NCD without involvement of alcohol neurotoxicity and with a characteristic ataxia of stance and gait exists and may be irreversible. We did not find any evidence for lasting cognitive abnormalities or a cerebellar cognitive-affective syndrome (CCAS) in this patient.

## 1. Introduction

Worldwide, approximately 10–20% of all people with alcohol use disorder develop a severe deficiency of vitamin B1 or thiamine in the second half of their life, due to a deficient diet and often because of thiamine loss through diarrhea and vomiting, in combination with an increased thiamine requirement for alcohol metabolization [1,2,3,4,5]. As a result, these patients develop Wernicke’s encephalopathy (WE), a severe neuropsychiatric disease that may be diagnosed on the basis of a characteristic triad of symptoms first described by Carl Wernicke: an ataxia of stance and gait, mental confusion, and eye movement disturbances (nystagmus, abducens paresis or conjugate gaze palsy) [4,5,6]. However, since the 1980s, we have known that only 16% of all WE patients present with this triad. Some 19% have no documented clinical signs, and the rest a rather atypical presentation with mental confusion only [7]. Consequently, WE is notoriously difficult to diagnose. In a large retrospective study on histopathological examinations, 80% of patients with histopathological evidence of WE did not receive this diagnosis during their lifetime [7]. The diagnosis is rarely missed in patients who die, who present with a complete triad, or who develop Korsakoff’s syndrome, a residual syndrome in patients whose WE is treated too late, characterized by a severe amnesia and other cognitive and behavioral symptoms [4,8]. However, in less severe forms of thiamine depletion, the resulting cognitive disorders are often less conspicuous and may be unjustly attributed to the direct toxic effects of alcohol [9,10,11]. This misattribution has long obscured the fact that alcohol abuse usually only provides the context in which thiamine depletion develops, and that alcohol neurotoxicity does not contribute to the development of KS or other residual syndromes in any essential way [5,8,12]. In line with this, WE and KS have been reported in cases with non-alcoholic thiamine deficiency, such as bariatric surgery [13], cancer [14], anorexia nervosa [15], or hyperemesis gravidarum [16]. At present, the most reliable clinical diagnosis of WE is achieved with the Caine criteria. At least two out of the following four criteria should be present in order to arrive at a definitive diagnosis: (1) dietary deficiencies, (2) oculomotor abnormalities, (3) cerebellar dysfunction, and (4) either an altered mental state or mild memory impairment [17].

Ataxia of stance and gait is one of the characteristic symptoms of WE [4,6,17]. However, it may also develop as an isolated symptom in people with alcohol use disorder, without other signs of WE. In these cases, the patients receive the diagnosis “alcoholic cerebellar degeneration” (ACD) [18,19]. These patients usually present with an ataxia of stance and gait, but without other definite cerebellar symptoms. Dysarthria and ataxia in the upper limbs are absent or only minimal. This restricted cerebellar syndrome develops on the basis of a restricted cerebellar atrophy. The anterior vermis and often the paramedian parts of the superior anterior lobes are affected, but no or only subtle damage is found in the posterior and lateral parts of the cerebellum. Microscopically, all layers of the cerebellar cortex in the affected areas are damaged, with narrowing of the molecular layer and patchy loss of granule cells, but a severe loss of Purkinje cells is most characteristic. Little or no atrophy or histopathology is found in the posterior vermis or in the posterior cerebellar lobes [4,20,21]. On CT and MRI, the characteristic distribution of the atrophy is clearly visible, especially in more pronounced forms of ACD [22,23]. ACD is not a rare disorder, probably even the most common form of chronic cerebellar ataxia [24,25], although the reported incidence and prevalence vary widely, mainly depending on geographical factors. Post-mortem studies in Norway have reported a prevalence between 26.8 and 41.9% in all people with alcohol use disorder [2,26,27], whereas in Japan, a prevalence of 3.2 to 10.9% was found [28,29]. 

Ataxia of stance and gait on the basis of a restricted cerebellar cortical atrophy has been described since 1900 [30], but its association with alcohol abuse became apparent only after 1931, and the term “alcoholic cerebellar degeneration” was not used before 1940 [18,31]. Most studies after 1940 identified alcohol toxicity as the obvious cause, but some authors suspected that vitamin deficiency might play a role as well. The landmark study by Victor et al. [32] that established ACD as a separate clinical entity appeared in 1959 [31]. All patients with ACD in that study were people with alcohol use disorder, but their detailed investigations had convinced the authors that it was not alcohol neurotoxicity but thiamine depletion that was the chief causative factor of the restricted cerebellar atrophy and the ataxia of stance and gait. That is, (1) six of their 50 patients had been abstinent from alcohol at the time that the cerebellar disease began, three of them because they were hospitalized and could not drink alcohol; (2) 75% of their sample was chronically malnourished, and nine patients had suffered a particularly marked weight loss over a short time before a sudden worsening of their cerebellar ataxia occurred; and (3) of the 11 autopsied cases, eight had WE-neuropathology or other thiamine-depletion related pathology. Thus, arguably, ACD might better be characterized as “nutritional cerebellar degeneration” (NCD) [30,32]. However, cases with non-alcoholic (i.e., purely nutritional) restricted cerebellar degeneration are rare [30], and animal studies have demonstrated that alcohol damages the cerebellum [33,34]. Even though several patient reports with NCD have been published over the years [30] and NCD can be induced in rats and other animals by a thiamine-depleted diet or a diet with pyrithiamine-induced thiamine deficiency (PTD) [35,36,37], the debate on the essential cause(s) of ACD is still ongoing [30,38,39]. While some authors found no atrophy of the cerebellar vermis in non-alcoholic WE and argue that it is exclusive to alcoholic WE [40], others have presented evidence that cerebellar involvement (hyperintense MRI signal on T2 and FLAIR) is, on the contrary, characteristic of non-alcoholic WE [41] or found it in both groups [42]. Moreover, it is unclear whether NCD may result in an irreversible restricted cerebellar degeneration with a persistent ataxia of stance and gait. The three previously published case studies on patients with NCD (and in all three cases, also WE) provide no evidence for irreversible damage, as two of these cases died within weeks of the diagnosis [43,44]. In the third patient, the duration of survival after WE and NCD onset was not reported [32].

Another unresolved issue is whether the restricted cerebellar degeneration of WE and ACD in itself results in cognitive impairments. Initially, the cerebellum was considered to be important for motor coordination only, but the current perspective on the cerebellum is based on the recognition that it also plays an important role in cognitive and emotional functions [45,46,47]. Consequently, cerebellar lesions are thought to result in cognitive and affective dysfunction as well, which is conceptualized in the cerebellar cognitive and affective syndrome (CCAS) [48,49,50].

Here, we present the unique case report of a patient who developed a pure (i.e., non-alcoholic) NCD and a WE after a COVID-19 infection, upon whom we were able to perform an in-depth neurological, neuroradiological, laboratory and neuropsychological assessment.

## 2. Materials and Methods

### 2.1. Patient

The patient is a 46-year old woman who was infected with the SARS-CoV-2 virus in the second wave of the pandemic, which resulted in COVID-19, verified by a polymerase chain reaction test. Her general medical condition worsened in the two months after the beginning of the infection, with protracted vomiting and her being unable to keep food down. She complained of nausea and a loss of taste and smell. Her husband took care of her, but found himself in a situation in which he was unable to provide the care and support that was required. As a result, a general practitioner did not become involved until late in the process, at which time the patient was already severely ill and undernourished. Before her COVID-19 infection, she weighed 89.3 kg (height: 164 cm, BMI: 33.2). Approximately three months after the onset of COVID-19, she was admitted to a general hospital. At that point, her weight loss amounted to 15.2 kg (weight: 74.1 kg, BMI: 27.6). When a home-care assistant finally managed to visit our patient, she found her to be very ill and called an ambulance. Upon admission to the hospital, the patient was very confused and restless and unable to stand on her feet or sit up straight. She complained about diplopia, but due to limited cooperation, a full neurological assessment was impossible. Her responses to questions were confused and inadequate, but most of the questions did not elicit any response. There was no nuchal rigidity, no muscular hypertonia, and no lateralization. A clinical diagnosis of nutritional WE was made, as she suffered from thiamine deficiency and presented with the classical picture of this disease: a complete triad of Wernicke [6] (here: abducens paresis, ataxia of stance and gait, mental confusion) and all four signs of the Caine criteria [17] (here: (1) dietary deficiencies, (2) oculomotor abnormalities, (3) cerebellar dysfunction, and (4) either an altered mental state or mild memory impairment), where only two are required for a clinical diagnosis of WE. Tube feeding was started, and the patient was treated with thiamine replacement. This began with 500 mg thiamine, intravenously, three times a day for three days, and was followed by 500 mg, orally, once a day. After two weeks, she received 50 mg orally twice daily, alongside all other necessary interventions, after which her medical and mental condition improved. After discharge, she was admitted at a long-stay rehabilitation unit to recover further. Four months after admission to the rehabilitation department, she was then admitted to our Korsakoff Center for a full diagnostic work-up. At that point, she was no longer confused and the ocular paresis (nervus abducens) had disappeared, but a severe ataxia of stance and gait and mild memory problems were still present. Before her COVID-19 infection, she had no coordination problems. There was neither a family history of spinocerebellar degeneration, nor of any other cerebellar disorder. Of note, the patient was a teetotaler, fully abstinent from using alcohol (she reportedly consumed an alcoholic beverage once when she was in her early 20s, but she did not appreciate the experience at all). In addition, she already received support at home from a case manager before the COVID-19 infection, because of her and her husband’s lower intellectual abilities. Before her illness, the patient was able to work in a sheltered environment and function independently in most domains of daily living. 

### 2.2. Neuropsychological Assessment

The cognitive assessment consisted of screening instruments that were administered during the course of her recovery and an extensive neuropsychological assessment (NPA) five months after admission to the hospital.

With respect to the screening instruments, the Montreal Cognitive Assessment (MoCA) including its Memory Index Score (MIS) [51] was administered during the intake (one months after the WE onset) and at the time of the NPA to assess her overall cognitive status. In addition, the Frontal Assessment Battery (FAB) was administered as an index of frontal executive symptoms [52]. Finally, the Schmahmann CCAS Scale [53] was administered as a screen for the CCAS as a syndrome.

The NPA consisted of a large battery of validated neuropsychological tests tapping the major cognitive domains intelligence, orientation, working and episodic memory, executive function, attention, language, visuoconstruction and social cognition. The difficulty level of the test battery was tailored to the lower intellectual ability level of the patient and also included measures of performance validity to ensure optimal effort (see Appendix A for an overview of the individual tests used and Lezak et al. [54] for more details with respect to test administration and scoring). All test outcomes were interpreted using the international consensus criteria for neuropsychological performance test scores (and summarized in the T score range per cognitive domain) [55]. Repeated cognitive assessments were analyzed with Reliable Change Index (RCI) analyses when possible. 

### 2.3. Standardized Observations

In addition to the NPA, observations in the domains of apathy, everyday cognitive performance and confabulations were made with the following observation scales: the Apathy Motivation Index (AMI) [56], Behavior Rating Inventory of Executive Function (BRIEF-A) [57], the Nurses’ Observation Scale for Cognitive Abilities (NOSCA) [58], and the Nijmegen–Venray Confabulation List (NVCL-R) [59]. All but the NOSCA were completed once by the professional caregiver who was responsible for the care of the patient, and who was able to observe her behavior throughout the clinical stay. The NOSCA was administered 13 times over a 9-day period (six times in the evening, and seven times during the daytime). 

### 2.4. Neurological Assessment

An extensive neurological examination was performed at our Korsakoff Centre four and a half months after admission to the hospital, and consisted of an examination of the patient’s mental status, cranial nerves, motor system, coordination, sensibility, and reflexes. 

### 2.5. Imaging

T1-weighted, T2-weighted, diffusion-weighted, and fluid-attenuated inversion recovery (FLAIR) T2-weighted magnetic resonance imaging (MRI) of the brain was performed as part of routine clinical assessment at three time points (3 days, one month, and seven months after hospital admission) on 1.5 Tesla scanners. All MR images were visually analyzed by an experienced radiologist and a neurologist. 

In addition, an X-ray of the thorax and CAT scans of brain, thorax and abdomen were performed.

### 2.6. Laboratory Investigations

The following tests were performed: hematocrit, hemoglobin, white cell count, differential count, platelet count, red cell count, mean corpuscular volume; urea, creatinine, estimated glomerular filtration rate, sodium, potassium, chloride, calcium, magnesium, phosphate, total protein, albumin, total bilirubin, direct bilirubin, alanine aminotransferase, aspartate aminotransferase, lactate dehydrogenase, alkaline phosphatase, gamma-glutamyl transferase, albumin, c-reactive protein, lipase, lactid acid, creatine kinase, glucose, hemoglobin A1c, (hepta-, hexa- and pentacarboxylic) porphyrins, coproporphyrin, vitamin B1 (thiamine), vitamin B6, vitamin B12, toxicological screening (all recreational drugs, lithium, digoxin, paracetamol, salicylic acid), blood cultures, treponema antibodies, HIV; standard blood gas analysis, standard urinalysis, standard liquor analysis, and in addition, NMDAR antibodies, enterovirus, herpes simplex virus, varicella zoster virus, and cytomegalovirus in liquor. 

## 3. Results

### 3.1. Clinical Course

Upon admission to the hospital, in the acute phase of the WE, the patient was in a state of severe mental confusion, with hallucinations and disinhibited behaviour. She did not recognize her own case manager, had uncoordinated eye movements and could not hold her balance, even in a sitting position. After a few months of clinical treatment, including thiamine supplementation, her medical and mental state had improved dramatically. Upon admission to our Korsakoff Center five months after the WE onset, her orientation was intact and she was able to engage in a normal conversation. The patient stayed at our Korsakoff Center for six weeks, during which time the clinical diagnostic testing was performed. Thiamine levels were normal and remained stable during the admission. The patient was intrinsically motivated for the diagnostic work-up, but she was also nervous about staying far from home and not having her husband and her regular caregivers nearby. Her homesickness persisted and she called her husband and case manager on a regular basis. During her stay in our center, a stressful period in her life, she was easily upset and soon became emotional. Professional caregivers provided emotional closeness and guidance, but were not able to substitute the patients’ regular caregivers and close relatives. In terms of daily functioning, the patient showed the ability to perform independently in her self-care and performed well on daily and therapeutic activities. She did not need reminders on her scheduled activities, and confabulations were not observed. Even though the patient did not have any memories of the acute phase of the WE (when she was very ill), she showed insight in her present functioning, impairments, and recovery. Furthermore, the patient received physical therapy to strengthen her physical condition and the muscle strength in her legs. She noticed a slight improvement in her ability to walk herself, but the use of a walker remained necessary indoors and outdoors. After good clinical recovery, the patient was discharged from our center six months after the onset of WE and retuned home, where her case manager provided more intensive outpatient support than before, due to the patients increased need for emotional safety. 

### 3.2. Neuropsychological Assessment

One month after the WE, the MoCA total score was 15/30, with an MIS of 5/15. Five months after the WE, the MoCA total score was 20/30, with an MIS of 5/15. The change in the patient’s overall cognitive performance was a statistically significant improvement (RCI = 2.09, *p* < 0.05, one-tailed), but her performance on the MoCA was still below the widely-used cut-off score of < 26 [51]. Her performance on the FAB was at maximum (18/18), and her score on the Schmahmann CCAS Scale was 73/120, with 6/10 failed tests, which was indicative of CCAS as a syndrome with a score above the cut-off score of < 2 failed tests. 

The results of the extensive neuropsychological assessment five months after the WE are presented in Table 1. The patient passed all performance validity tests, indicating that she displayed optimal effort during testing. The performance on the neuropsychological assessment was not hampered by any visual, auditory, language or motor problems. Assessment of her intellectual abilities revealed performances in the mild intellectual disability range (Full-Scale IQ = 61), which was in agreement with her premorbid level of intellectual functioning and adaptive functioning. She was optimally oriented in time and place. Her performance on tests of episodic memory varied from exceptionally low on a story recall test to average on word list learning, object–location memory and memory for visual associations, a performance which was deemed in line with her pre-morbid functioning (and not indicative of an amnestic disorder). Her working memory performance was in the average range. In the domain of executive function, she performed, in general, very well (in the average range), with below-average performances on higher-order abstract planning tasks, again in line with her premorbid intellectual functioning. In the domain of attention, her performance was below average, and her language performance was in the average range. On a test for visuo-construction, her performance was exceptionally low, and her social cognitive abilities were in the below-average range. 

In all, the results of the cognitive screenings showed cognitive improvement between 1 to 5 months after the WE. The outcome of the neuropsychological assessment was deemed in line with her premorbid intellectual functioning, with weaknesses in the visuo-construction and abstract/executive domains, but without specific cognitive impairments that could be attributed to her illness or the WE. In the DSM-5-TR classification [60], she thus did not meet the criteria for a neurocognitive disorder. 

### 3.3. Standardized Observations

On the AMI, the observer-rated scores for behavioral (1.50), social (1.67) and emotional (0.50) apathy were below the established cut-off scores. On the BRIEF-A, all subscales were in the normal range (T-scores between 40 and 60), apart from the subscale of emotion regulation, which was elevated (T = 69); this was indicative of the patient’s observed problems in emotion regulation. Observations with the NOSCA resulted in a maximum score for the subscale Consciousness, reflecting a fully responsive and alert state. The mean total score of the repeated observations of the other domains was 18.0 (SD = 0.9), indicating only minimal observed difficulty in cognitive activities. On the NVCL-R, no spontaneous confabulations were observed (0/24), there were only minimal evoked confabulations (4/28), and there were no problems in reality monitoring (0/8), with all scores below the established cut-off scores. 

### 3.4. Neurological Examination

Mental Status: At the time of the neurological examination (five months after admission to the hospital), the patient was alert and oriented to person, place, and time with normal speech. The memory was mildly disturbed; the thought process was intact. Cranial Nerves: I-XII: no abnormalities. Extraocular movements were intact without ptosis. The abducens paresis that was observed in the acute phase was no longer present. Her voice was normal, and there were no signs of any form of dysarthria. Motor system: The patient had a slightly decreased muscle strength, probably due to a residual weakness after cachexia and prolonged immobilization. She had a mild, bilateral, relatively slow postural tremor in both arms and hands (reportedly present since early childhood). She had a left peroneal nerve paresis with ankle foot orthesis on the left side (a residual disorder after two operations for a complicated L5-S1 intervertebral disc herniation left). Coordination: She performed the finger-to-nose test normally bilaterally and was able to rapidly perform alternating movements. She had a severe ataxia of stance and gait, necessitating a rollator. Sensation: Her sensation was intact to pain and light touch bilaterally, only disturbed in the left dermatomes L5-S1. Reflexes were normal and symmetrical, except from a lowered Achilles reflex on the left. 

### 3.5. Imaging

A thorax X-ray and CAT scans of brain, thorax and abdomen did not show any abnormalities. Magnetic resonance imaging on the fourth day after hospital admission showed hyperintensities on FLAIR sequences that are typical of active Wernicke’s encephalopathy (see Figure 1 and Figure 2). Their genesis is not completely understood. Possibly, thiamine deficiency causes an inability to sustain osmotic gradients of cell membranes and a failure to maintain membrane integrity, leading to intra- and extra-cellular edema. On all MRIs, there was anterior vermal atrophy characteristic of ACD and NCD (see Figure 3). The subacute development in the context of definitely established thiamine depletion makes a neurodegenerative origin highly unlikely.

### 3.6. Laboratory Investigations

On hospital admission, our patient had mild electrolyte disturbances and a slightly increased hematocrit and hemoglobin due to dehydration. She also suffered from respiratory alkalosis (due to hyperventilation) and metabolic acidosis, with an elevated anion gap and an elevated lactic acid. Her liver enzymes were also slightly elevated. All values normalized within a few days with standard treatment. Our patient had a clear deficiency of thiamine (26 nmol/L; normal range: 83–187 nmol/L), for which she received thiamine suppletion. Levels of vitamins B12 and B6 were within the normal range (B12: 555 pmol/L, normal range:156–672; B6: 103 nmol/L; normal range: 35–110). Shortly after the start of tube feeding, she developed a refeeding syndrome with lowered sodium, potassium, magnesium and phosphate, which disappeared within days with standard treatment. There was clear evidence that our patient had suffered from a COVID-19 infection, but it was no longer active. All other laboratory assessments, including those of urine and liquor, were within the normal range or with minimal, insignificant and short-lived deviations.

## 4. Discussion

The results of the diagnostic workup, completed five months after admission to the hospital, showed that the initial nutritional WE of our patient had completely recovered. That is, the patient was fully oriented and alert, had excellent illness insight and did not show any signs of confabulation behavior. The performance on the MoCA showed improvement over time, in line with her general clinical recovery. The results of the extensive neuropsychological assessment showed performances that were in agreement with her premorbid level of cognitive functioning (i.e., lower intellectual ability), and did not show impairments in the cognitive domains. Episodic memory was a strength within her cognitive profile, and visuo-construction a weakness. Thus, the patient did not meet the criteria for Korsakoff’s syndrome according to DSM-5-TR [60] (alcohol-related major neurocognitive disorder, amnestic/confabulatory type) or the criteria outlined in Kopelman [61]. 

Our patient also did not fulfill the criteria for the CCAS or “Schmahmann syndrome” [62], as there were no cognitive impairments, including none in the social-cognitive domain. The observed slight problems in emotion regulation were thus not accompanied by a deficit in social cognition, and were too subtle for meeting the CCAS criteria. This is an important observation. Traditionally, the cerebellum is thought to be responsible for motor coordination only. However, in 1888, Gowers did suggest that it might be involved in cognitive functions as well, but his conjectures did not take hold at that time [63,64]. Schmahmann was the first who convincingly argued and demonstrated that the cerebellum does not exclusively subserve motor functions, but is also involved in emotion and cognition [62,65]. Initially, his ideas did not receive universal support [66,67], but they have become widely accepted in recent years [46,48,49,68,69]. 

It has been demonstrated that in several diseases of the cerebellum, not only motor coordination but also emotions and several cognitive functions may be disturbed [70,71,72]. Schmahmann developed his ideas on the non-motor effects of cerebellar dysfunction into the concept of CCAS. Since then, the CCAS has been studied in hereditary spinocerebellar ataxias, in vascular cerebellar syndromes and in many other cerebellar diseases [71,72]. However, to date, there is only one study on CCAS in ACD, published by Fitzpatrick et al. [73], which reported on cognitive and emotional deficits in 49 people with chronic alcohol use disorder and compared the findings to the results in 29 matched healthy controls, and aimed to assess whether the severity of the cognitive and emotional deficits is linked to the severity of the ataxia (“the clinical signs of ACD”) in the people with alcohol use disorder. They found “evidence of a relationship between the degree of ACD and some areas of cognitive and emotional functioning, including language, executive functioning, processing speed and affect processing” (p. 529), and concluded that their results “provide supportive evidence for a cerebellar contribution to some of the cognitive and emotional deficits that occur in chronic alcoholics” (p. 530). The authors argue that these findings are in line with Schmahmann’s dysmetria of thought hypothesis.

While Fitzpatrick et al. [73] assume that the observed association between ataxia and cognitive deficits more or less proves that ACD is responsible for both, the cognitive deficits that are associated with ACD in their study are possibly better explained by concomitant thiamine deficiency-induced thalamic damage [5,8,9,10,11,74]. This alternative explanation is also in complete agreement with Schmahmann’s latest hypotheses [45,47] and the results of experimental studies [49,75]. It has been demonstrated that the anterior cerebellum (the part that is damaged in ACD) chiefly contributes to motor coordination and emotional functions, but not to cognitive functions [47]. It is the posterior cerebellum—the part that is usually completely spared in ACD—which is argued to contribute significantly to cognitive functions [47], with posterior cerebellar damage resulting in cognitive deficits [45]. 

In our case, the concomitant thalamic damage was obviously mild and partly reversible, resulting in transient cognitive dysfunction, but the anterior cerebellar damage was clearly considerable and irreversible. Consequently, the resulting ataxia is still severe more than a year after the onset of WE, corroborating the current perspective on CCAS and its topography [47]. Therefore, the results of our case study seriously undermine the hypothesis that ACD may lead to a CCAS [73,76]. It also demonstrates—in full agreement with current evidence on cerebellar contributions to cognition [45,46,47,48,49,50]—that at present, there is no compelling reason to entertain the hypothesis that thiamine depletion-induced cerebellar damage contributes to cognitive deficits after WE or in KS [77].

Our case is also a further illustration that COVID-19 may sometimes lead to WE. It is conceivable that COVID-19 is a direct causative factor in the development of cerebellar degeneration, for it may lead to several neurological complications, including encephalopathies [78,79,80]. However, these encephalopathies are non-specific and usually have a good prognosis, and only very rarely has COVID been associated with ataxia [81,82]. Moreover, in the typical cases of WE after COVID-19, this infection apparently provided only the context in which thiamine depletion could develop, i.e., through malnutrition after persistent nausea and vomiting or after dysgeusia [83,84,85,86]. Furthermore, patients on an ICU with sepsis or in catabolic states are always at risk of thiamine depletion, with or without COVID-19, but they usually respond with immediate neurologic improvement when treated with intravenous thiamine [87,88]. It is conjectured only in some atypical WE-cases, with additional neurologic disorders or other co-morbidities, that there may have been some direct influence of COVID-19 or associated immunologic mechanisms [89,90].

Our patient had a pre-existing mild intellectual disability, but was able to function independently with only minimal support. Her post-WE neuropsychological test scores were found to be in line with her pre-existing cognitive strengths and weaknesses, and did not meet the criteria for a (mild) neurocognitive disorder. Our findings also show that caution is required when administering and interpreting cognitive screens, such as the MoCA or CCAS scale, in individuals with lower intellectual abilities or lower education levels, as such screening instruments have a suboptimal specificity that is attenuated in individuals with lower premorbid cognitive function. For instance, there is abundant evidence that a substantial proportion of cognitively healthy participants are misclassified as having cognitive impairment when using the established cut-off score of the MoCA, as this cut-off score insufficiently adjusts for (lower) educational attainment [91]. Similarly, a recent study using the CCAS scale for detecting cognitive deficits in spinocerebellar ataxia type 3 also demonstrated that some of the healthy controls performed in the impaired range on this scale [92]. Thus, our study highlights the need for thorough neuropsychological assessment in patients suspected of having cognitive deficits on cognitive screens, and the need to relate the outcome of such an assessment to the premorbid level of cognitive and everyday functioning.

## 5. Conclusions

We presented a case study of a 46-year-old woman with NCD and WE after severe COVID-19, but without an acute respiratory distress syndrome. She was a teetotaler, and had consumed only one alcoholic beverage in her entire life, which excludes any contribution of alcohol neurotoxicity to her condition. This is a unique case because it is the first and, so far, only case with a completely documented and fully conclusive demonstration that without any involvement of alcohol toxicity, thiamine depletion can lead to a characteristic restricted form of cerebellar degeneration, with a severe and persistent ataxia of stance and gate, without dysarthria, without intention tremor and without ataxia in the upper limbs. With this case, we have provided strong evidence that alcohol neurotoxicity is not a necessary causative factor for the development of a characteristic, severe and persistent restricted cerebellar degeneration. The existence of NCD as a separate clinical entity in humans, caused by thiamine depletion alone, is thus definitely established now. 

Our case is also unique because it is to our knowledge the only documented case of pure NCD in which the presence of cognitive disorders was assessed by an extended neuropsychological test battery, standardized clinical observations and questionnaires, including the CCAS-scale. The absence of de novo cognitive impairments and the absence of a CCAS in our patient is in full accord with the current conception of the topical organization of the cerebellum [45,47]. According to this conception, which is supported by an impressive body of evidence [45,46,47,48,49,50], the anterior vermis subserves motor control in the lower limbs and also subserves emotion (it is the “limbic cerebellum”, according to Schmahmann [93]). Our patient showed emotional instability, which appears to continue up until today, but it is difficult to decide whether this is best explained through her difficult circumstances or through anterior vermis damage. 

## Figures and Tables

**Figure 1 jcm-12-02511-f001:**
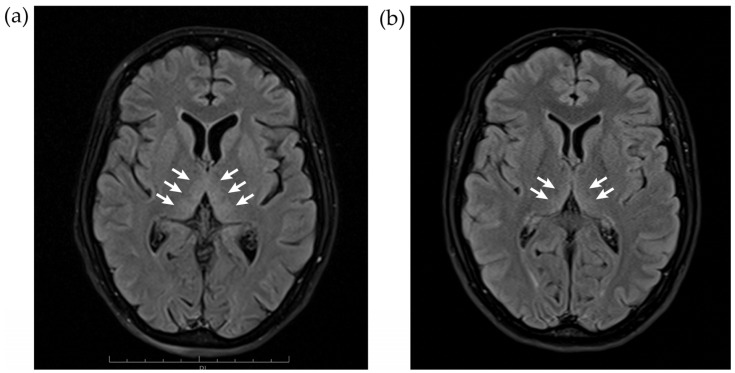
Axial 1.5 Tesla T2 fluid-attenuated inversion recovery (FLAIR) weighted MRI scans showing the bilateral thalamic hyperintensities (arrows) during the stage with acute Wernicke encephalopathy signs, four days after hospital admission (echo time [TE] = 78 ms; repetition time [TR] = 5070 ms) (**a**), which were significantly reduced at the time of the neuropsychological assessment five months later (TE = 84 ms; TR = 9000 ms) (**b**).

**Figure 2 jcm-12-02511-f002:**
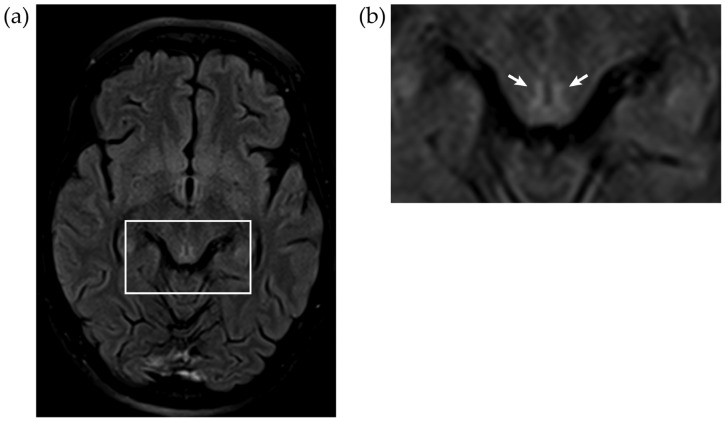
(**a**) Axial 1.5 Tesla T2 fluid-attenuated inversion recovery (FLAIR) weighted MRI scan showing the (**b**) hyperintensities of the periaqueductal grey (arrows) during the stage with acute Wernicke encephalopathy signs, four days after hospital admission (TE = 78 ms; TR = 5070 ms).

**Figure 3 jcm-12-02511-f003:**
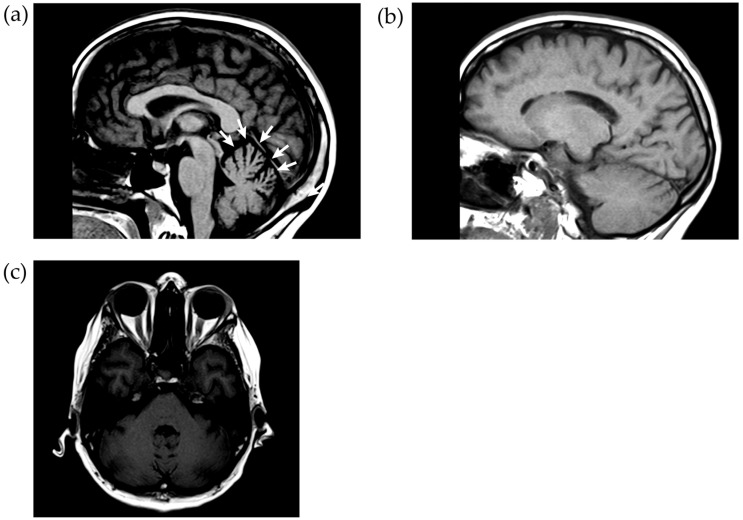
Sagittal 1.5 Tesla T1 weighted MRI scan made 1 month after hospital admission, showing (**a**) the anterior vermal atrophy in the mid-sagittal plane (arrows), (**b**) normal cerebellar hemispheres in a more lateral plane (TE = 8.7 ms; TR = 400 ms), and (**c**) a horizontal section showing that the cerebellar hemispheres are intact (TE = 8.7 ms; TR = 510 ms).

**Table 1 jcm-12-02511-t001:** Overview of the patient’s performance on neuropsychological assessments per cognitive domain (standard scores and classification).

Cognitive Domain	T-Score Range ^1^	Classification ^2^
Intelligence	34–40	Mild intellectual disability
Orientation	-	Unimpaired
Episodic memory	28–73	Average
Working memory	52–61	(High) average
Executive function	20–67	(Below) average
Attention	30–48	Below average
Language	39–50	Average
Visuo-construction	26	Exceptionally low
Social cognition	29–37	Below average

^1^ T-scores have a mean of 50 and an SD of 10; ^2^ Classification based on the labels reported in Guilmette et al. [55].

## Data Availability

Not applicable.

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
