# Peer review of "Coordination and Cognition in Pure Nutritional Wernicke’s Encephalopathy with Cerebellar Degeneration after COVID-19 Infection: A Unique Case Report"

_jcm, 2023, doi:10.3390/jcm12072511_

Round 1

Reviewer 1 Report

We would like to thank you for submitting this paper to our special issue on Wernicke's encephalopathy. As a reviewer, I have read a very unique case report of Wernicke's encephalopathy associated with malnutrition and cerebellar degeneration following COVID-19 infection. This is a single case report and the level of clinical evidence is poor, as there are virtually no similar reports in the past. However, it is almost certainly a clinically very interesting report. I would also rate the MRI images presented as qualified.

………………

<Major 1> In general, Wernicke's encephalopathy is a chronic encephalopathy with a subacute onset that is difficult to diagnose. The introduction describes the circumstances leading to the diagnosis of Wernicke's encephalopathy. However, there is no clear description of the diagnostic criteria. Currently reliable diagnostic criteria should be noted in the Introduction first.

<Major 2> The case history states that the patient lost 15 kg of body weight in the first two months after the COVID-19 infection began. I understand that she has lost weight, but there is no mention of the Kg value of her original weight and her weight after the loss. Please be specific and add your height and weight. Also, please describe how the BMI value changed with the weight loss after the coronary infection.

<Major 3> There is a description of increased vomiting in response to weight loss after a new coronavirus infection. Post-infection complications of this infection are commonly known as olfactory and taste disorders. In her example, was there any impairment of taste or smell? Did she have any picky eaters or changes in food preferences?

<Major 4> Are the cerebellar findings in her MRI images presented atrophic or degenerative? I believe that in order to perform an imaging evaluation of her cerebellum, it is necessary to present not only a sagittal section but also a horizontal section. Also, were there any findings of spinal cord atrophy as a complication?

<Major 5> The basis for the diagnosis of encephalopathy is evidenced by the results of an electroencephalogram (EEG) examination. An EEG for encephalopathy will show high amplitude slow waves and flattened findings. Sometimes epileptic seizures may also occur. By the way, have you done several EEG studies on her?

………………

<Minor 1> It is noted that she had normal blood tests for vitamins other than thiamine. I would like to add the type of vitamin test that was specifically examined.

<Minor 2> Has a spinal fluid test been performed?

<Minor 3> Is there a family history of spinocerebellar degeneration?

<Minor 4> You mention that thiamine levels were low, but how low does Wernicke's encephalopathy generally occur?

The authors need to respond sincerely to some of my comments. I look forward to your written responses.

………………

Best regards,

Dr. Reviewer

Author Response

Reviewer #1:

We would like to thank you for submitting this paper to our special issue on Wernicke's encephalopathy. As a reviewer, I have read a very unique case report of Wernicke's encephalopathy associated with malnutrition and cerebellar degeneration following COVID-19 infection. This is a single case report and the level of clinical evidence is poor, as there are virtually no similar reports in the past. However, it is almost certainly a clinically very interesting report. I would also rate the MRI images presented as qualified.

We thank the reviewer for the very positive comments on our manuscript.

<Major 1> In general, Wernicke's encephalopathy is a chronic encephalopathy with a subacute onset that is difficult to diagnose. The introduction describes the circumstances leading to the diagnosis of Wernicke's encephalopathy. However, there is no clear description of the diagnostic criteria. Currently reliable diagnostic criteria should be noted in the Introduction first. We fully agree that these criteria are helpful to the reader. Therefore, we have added the consensus diagnostic criteria (i.e. proposed by Caine) to the Introduction in the revised version (see lines 57-60).

<Major 2> The case history states that the patient lost 15 kg of body weight in the first two months after the COVID-19 infection began. I understand that she has lost weight, but there is no mention of the Kg value of her original weight and her weight after the loss. Please be specific and add your height and weight. Also, please describe how the BMI value changed with the weight loss after the coronary infection. We have added all the requested values in lines 130-133.

<Major 3> There is a description of increased vomiting in response to weight loss after a new coronavirus infection. Post-infection complications of this infection are commonly known as olfactory and taste disorders. In her example, was there any impairment of taste or smell? Did she have any picky eaters or changes in food preferences? Our patient complained of nausea and a loss of taste and smell. We have added this information to the case description (see line 127).

<Major 4> Are the cerebellar findings in her MRI images presented atrophic or degenerative? The MRI images were characteristic for the restricted cerebellar cortical degeneration that is described in the literature on alcoholic cerebellar degeneration and nutritional cerebellar degeneration. Its subacute development in the context of – definitely established – thiamine depletion excludes a neurodegenerative origin. We have added this to the Results section on neuroimaging (lines 312-313).

I believe that in order to perform an imaging evaluation of her cerebellum, it is necessary to present not only a sagittal section but also a horizontal section. Also, were there any findings of spinal cord atrophy as a complication? We have added a horizontal section to Figure 3 (c), and mention in the caption that this shows that both cerebellar hemispheres are intact (line 326). There were no findings of spinal cord atrophy as a complication.

<Major 5> The basis for the diagnosis of encephalopathy is evidenced by the results of an electroencephalogram (EEG) examination. An EEG for encephalopathy will show high amplitude slow waves and flattened findings. Sometimes epileptic seizures may also occur. By the way, have you done several EEG studies on her? No EEG studies were performed, but all other neuroimaging and laboratory assessments were very thorough and comprehensive and substantiated the diagnosis of Wernicke’s encephalopathy in accordance with the Caine criteria. There were no clinical signs of epileptic seizures.

<Minor 1> It is noted that she had normal blood tests for vitamins other than thiamine. I would like to add the type of vitamin test that was specifically examined. The normal values of B6 and B12 have been added. The hospital reports do not mention B2 and B3. We have added these details to the Results (lines 333-335)

<Minor 2> Has a spinal fluid test been performed? A cerebrospinal fluid test has been performed; this was already reported in section 2 (lines 217-219). The (negative) results of liquor and urine assessments have now been added to 3.6 (line 339).

<Minor 3> Is there a family history of spinocerebellar degeneration? There was neither a family history of spinocerebellar degeneration, nor of any other cerebellar disorder. We have added this information to the section “2. Materials and Methods – Patient” (lines 155-156).

<Minor 4> You mention that thiamine levels were low, but how low does Wernicke's encephalopathy generally occur? This a very good question but unfortunately, we cannot provide a conclusive answer, as there is no definite information on this topic in the scientific literature, as far as we know. The diagnosis of Wernicke's encephalopathy is missed more often than not. When the diagnosis is suspected, immediate treatment with thiamine usually follows, which is adequate and often lifesaving. The therapeutic window of thiamine is enormous. Laboratory assessments of thiamine are time-consuming and therefore often skipped when the diagnosis of WE has been made on clinical grounds (cf. the Caine criteria). 

Reviewer 2 Report

The authors describe a patient with a Wernicke's encephalopathy in the context of thiamine deficiency without significant alcholol consumption.  

THe case is labelled as 'unique', but it is not clear which aspects of the case the authors consider to be unique.  We are told that the patient's vomiting began shortly after Covid-19 infection, but we are not told why she developed this symptom.  

Nonetheless, the diagnosis of Wernicke's encephalopathy appears to be sound and the patient improved on dietary treatment.  The detailed neuroradiology, psychological assessments and documented absence of cerebellar cognitive affective syndrome is interesting.

I think it would be valuable to have more detail about the patient's clinical examination when she was admitted to hospital, which hopefully will be readily available.  The summary provided is very limited

Author Response

Reviewer 2:

The authors describe a patient with a Wernicke's encephalopathy in the context of thiamine deficiency without significant alcohol consumption.  

The case is labelled as 'unique', but it is not clear which aspects of the case the authors consider to be unique.  We apologize that this was not fully clear in our initial version. In the revised conclusion, we have changed the wordings and made the unique selling point of our case report more explicit (see lines 436-440 and 445-448).

We are told that the patient's vomiting began shortly after Covid-19 infection, but we are not told why she developed this symptom. Probably, this was because of a COVID-19 induced nausea and a loss of taste and smell. We added this information to the section “2. Materials and Methods – Patient” (line 127).

Nonetheless, the diagnosis of Wernicke's encephalopathy appears to be sound and the patient improved on dietary treatment.  The detailed neuroradiology, psychological assessments and documented absence of cerebellar cognitive affective syndrome is interesting. We thank the reviewer for these positive comments.

I think it would be valuable to have more detail about the patient's clinical examination when she was admitted to hospital, which hopefully will be readily available.  The summary provided is very limited. We have added additional information with regard to the patient's clinical examination when she was admitted to hospital (see 133-139 and lines 146-148).

Reviewer 3 Report

The authors present a clinical case of Wernicke's encephalopathy with nutritional cerebellar degeneration after a severe COVID-19 infection in an abstemious patient. The presentation is correct and comprehensive, detailing the clinical, analytical, radiological and psychiatric evolution. 

As a minor comment, the doses and route of administration of thiamine are not indicated.

If this journal accepts clinical cases, I believe this case is a good candidate for publication.

Author Response

Reviewer 3:

The authors present a clinical case of Wernicke's encephalopathy with nutritional cerebellar degeneration after a severe COVID-19 infection in an abstemious patient. The presentation is correct and comprehensive, detailing the clinical, analytical, radiological and psychiatric evolution. 

We thank the reviewer for these positive comments.

As a minor comment, the doses and route of administration of thiamine are not indicated.

We have added these details to the patient description (lines 146-147): This started with 500 mg thiamine intravenously three times a day during three days, followed by 500 mg per os once a day. After two weeks, she received 50 mg per os twice daily.

If this journal accepts clinical cases, I believe this case is a good candidate for publication

Thank you very much for this recommendation.

Round 2

Reviewer 1 Report

Dear Author and co-author, 

The reviewers reviewed a detailed case report on Wernicke's encephalopathy resubmitted by the author. The new paper has been accurately revised to address almost all of the reviewers' comments. The reviewers found the report to be more accurate, including the addition of the patient's BMI, the amount of drug administered, and the reference values. The reviewers found the paper to be of great interest to readers, even though it is a case report of only one patient.

The reviewer did not comment on the initial peer review, but let the reviewer review the entire document again and request additional comments on the MRI.

MRI FLAIR images show pale high-signal areas in the bilateral thalamus that improve after vitamin B12 treatment. Please add a note on the FLAIR condition of the MRI taken to address the question of how this shading should be evaluated.

For example, was the MRI machine used for this imaging a 1.0 Tesla machine, a 1.5 Tesla machine, or a 3.0 Tesla machine?

Also, in the spin echo (SE) imaging, how many milliseconds were the echo time (ET) and repetition time (RT) set?

The reviewer will require that you describe the exact imaging conditions for these MRI FLAIRs.

Thank you very much for submitting this very interesting paper.

Finally, the authors ask that you please consider the opinions of the other reviewers and make a comprehensive judgment.

Best regards

Dr. Reviewer

Author Response

Response to the reviewers

The reviewers reviewed a detailed case report on Wernicke's encephalopathy resubmitted by the author. The new paper has been accurately revised to address almost all of the reviewers' comments. The reviewers found the report to be more accurate, including the addition of the patient's BMI, the amount of drug administered, and the reference values. The reviewers found the paper to be of great interest to readers, even though it is a case report of only one patient.

We thank the reviewers for these positive comments. We have highlighted our changes in green in the manuscript document.

The reviewer did not comment on the initial peer review, but let the reviewer review the entire document again and request additional comments on the MRI.

MRI FLAIR images show pale high-signal areas in the bilateral thalamus that improve after vitamin B12 treatment. Please add a note on the FLAIR condition of the MRI taken to address the question of how this shading should be evaluated.

In lines 310-313, we added information on how to evaluate and interpret the MR findings, that is, the MR shows “hyperintensities on FLAIR sequences that are typical for active Wernicke’s encephalopathy (see Figures 1 and 2). Their genesis is not completely understood. Possibly, thiamine deficiency causes an inability to sustain osmotic gradients of cell membranes and a failure to maintain membrane integrity, leading to intra- and extra-cellular edema.

For example, was the MRI machine used for this imaging a 1.0 Tesla machine, a 1.5 Tesla machine, or a 3.0 Tesla machine?

The field strength of the MR scanners was already reported in the Methods section of our initial version of the manuscript (line 199, i.e. 1.5 T), but we have also added this to the MRI figure captions to clarify (lines 321, 324, and 328)

Also, in the spin echo (SE) imaging, how many milliseconds were the echo time (ET) and repetition time (RT) set? The reviewer will require that you describe the exact imaging conditions for these MRI FLAIRs.

In addition to the field strength, we have added the echo times (TE) and repetition times (TR) to the MR figure captions, see lines 320, 325, and 331.